# Transient and Dynamic Simulation of the Fluid Flow through Five-Way Electric Coolant Control Valve of a 100 kW Fuel Cell Vehicle by CFD with Moving Grid Technique

**Soo-Jin Jeong [1],* , Ji-hoon Kang [2], Seong-Joon Moon [1] and Gum-su Lee [2]**

1   Alternative Fuel Power System R&D Department, Korea Automotive Technology Institute, Cheonan-si 31214, Republic of Korea; sjmoon@katech.re.kr
2   R&D Center, INZI CONTROLS Co., Ltd., Gunjachoen-ro 171, Siheung-si 15090, Republic of Korea; inzijhkang@inzi.co.kr (J.-h.K.); leegumsu@inzi.co.kr (G.-s.L.)
*   Correspondence: sjjeong@katech.re.kr; Tel.: +82-41-559-3059

**Abstract:** In order to maintain the performance of a fuel cell vehicle, it is essential to maintain a constant temperature of the stack. Therefore, it is very important to distribute the optimal coolant flow rate to each major component under very diverse and rapidly changing dynamic operating conditions. The part responsible for this is a five-way electric coolant valve. Therefore, this study aims to investigate transient dynamic flow characteristics of the fluid flow through a five-way electric coolant valve (PCCV: Penta-Control Coolant Valve). To achieve this goal, this paper attempts a three-dimensional dynamic simulation of the fluid flow through the valve using a commercial CFD solver with moving mesh technique to consider flow inertia and dynamic flow in the opening and closing stages of the ball valve rotating motion. The dynamic flow characteristics and the thermal mixing inside the PCCV ball valve during the opening and closing stages are analyzed. It was found that the discrepancies between dynamic and steady-state simulations are remarkable when fluxes with different levels of enthalpy and momentum flow into the PCCV, leading to strong flow interference and flow inertia, while the discrepancies are relatively small at low rotation speed and weak flow interference. Subsequently, the effect of the dynamic flow characteristics of the valve on the dynamic thermal mixing characteristics at two different ball valve rotation speeds and rotation directions are investigated. It was found that the dynamic flow and thermal mixing characteristics inside the PCCV are greatly affected by the rotation speed, rotation direction, and degree of flow interference between fluxes. It also helps design better coolant control strategies and improves the FCEV thermal management system.

**Keywords:** five-way electric coolant control valve (PCCV: Penta-Control Coolant Valve); fuel cell vehicle (FCEV); thermal management system; computational fluid dynamics; moving grid technique; flow inertia; dynamic flow characteristics

## 1. Introduction

While the development of eco-friendly vehicles is emerging as a great alternative to solving global warming and the future depletion of fossil fuels, fuel cell vehicles are being considered as the final development stage for eco-friendly vehicles. Fuel cell vehicles are equipped with a fuel cell powertrain instead of a conventional gasoline/diesel engine. Major components include a fuel cell stack, driver system, electric power system, and control system. Among these, the operation system is an essential factor for stack operation and consists of a thermal management system, an air supply system [1], and a hydrogen supply system [2–4]. The thermal management system (TMS) includes a water pump that circulates a coolant, a three-way valve that determines the flow path according to the coolant temperature, a coolant ion filter to maintain coolant electrical conductivity, an

indoor air conditioner, a cooling module, and a Cathode Oxygen Depletion (COD) heater to improve stack cold start and durability.

The coolant temperature conditions in fuel cell vehicles are different, with the stack cooling below 80 °C and the electric drive cooling below 65 °C separating each cooling system. Compared to the coolant temperature condition of a conventional internal combustion engine approximately below 120 °C, the heat generation amount that is a reaction by-product is lower than that of the existing internal combustion engine. This leads to an increase in the size of the radiator.

For fuel cell vehicles, power performance, such as the rated output of the electric drive system, maximum vehicle speed, and hill-climbing ability, is determined by cooling performance, and the lifespan of the inverter's internal capacitor can also be determined. Therefore, the distribution of the cooling flow rate to each component included in the cooling system and the control of coolant temperature are critical design technologies creating a fuel cell vehicle. In this context, the coolant control ball valve, installed in the coolant circuit of FCEV, is one of the essential coolant control devices in fuel cell vehicles, allowing accurate control of the flow rate and direction of coolant [5].

Previously, the multi-way valve commonly used to control coolant flow rate and direction in the cooling circuit of a fuel cell vehicle was a three-way electrical coolant valve [6] or thermostat [7]. Figure 1 shows an example of a cooling circuit for a fuel cell vehicle. It shows that the thermal management system loop bypasses the radiator when the coolant temperature is low. When the coolant temperature is high, it cools through the radiator. In addition, a coolant always flows through the branch loop that passes through the air conditioning heater and ion filter and is designed to heat the internal passenger compartment and maintain the electrical conductivity of the coolant.

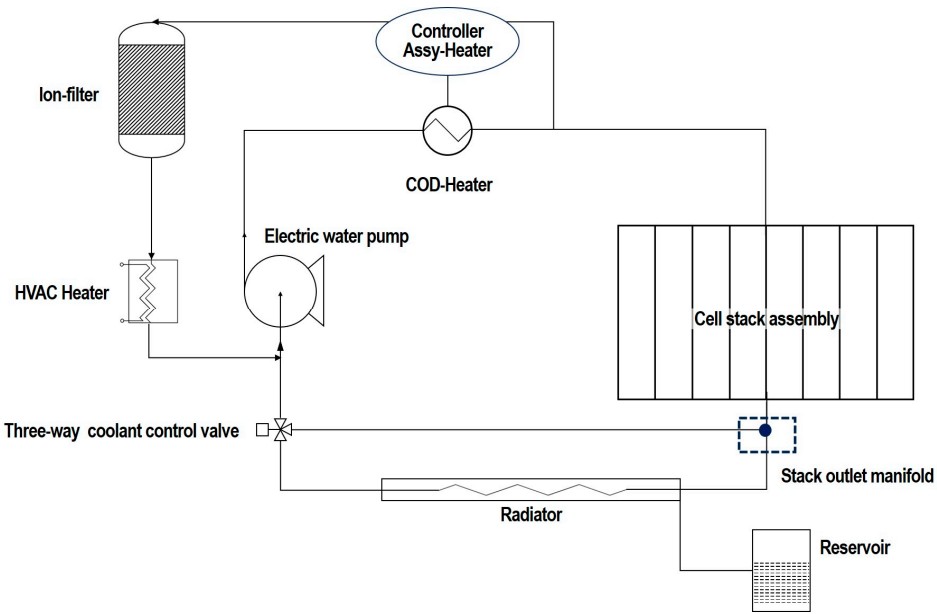

**Figure 1.** An example of a thermal management system for a fuel cell vehicle.

While an electric water pump or a thermostat can only control the entire coolant circulation or the coolant flow through a specific flow path, an electric coolant flow control valve (three-way valve), which has one inlet and two outlets, can control the coolant flows in multiple paths of a cooling circuit with a single unit. Since the electric coolant valve interconnects the stack water jacket and coolant flow paths for various heat exchangers, such as the radiator, air conditioning heater, and COD heater, the valve will be able to open up or shut off the coolant flow to the individual heat exchanging components, corresponding to a vehicle's operating condition. Furthermore, the valve can control the flow to each component, enabling a balanced distribution of the coolant flow between

the heat source and sink components for effective thermal management [8]. Coolant flow modulation is carried out by controlling the ball valve position in the coolant control valve using an electric motor attached to the valve assembly.

The cooling circuit of the early TMS, as shown in Figure 1, had the problem of a sharp decrease in fuel efficiency due to high power consumption in the cold start period. Therefore, to improve fuel efficiency, TMS with a COD heater was introduced, but problems with reduced fuel efficiency during starting and stopping and the durability of the stack and TMS still remain.

To compensate for these shortcomings, more precise control of the coolant flow rate became necessary and, as a result, an increase in the flow path became inevitable. In addition, it has become urgent to develop new products that can reduce the unit cost of existing systems, make them lighter, and overcome space limitations. In particular, the thermal management system of a high-performance 100 kW fuel cell operates according to a control strategy for each component, which makes it challenging to maximize the performance and lifespan of the fuel cell. To solve this problem, each component's hardware integration of the thermal management system was required. In addition, modularization and an integrated module control strategy in software were needed. Therefore, it was necessary to develop technology to secure market competitiveness through weight reduction and volume reduction by developing a product that integrated one three-way valve and one stack outlet manifold per vehicle in existing FCEVs into one structure. As shown in Figure 2, a five-way electric coolant control valve has been developed in this study to achieve the solutions mentioned above.

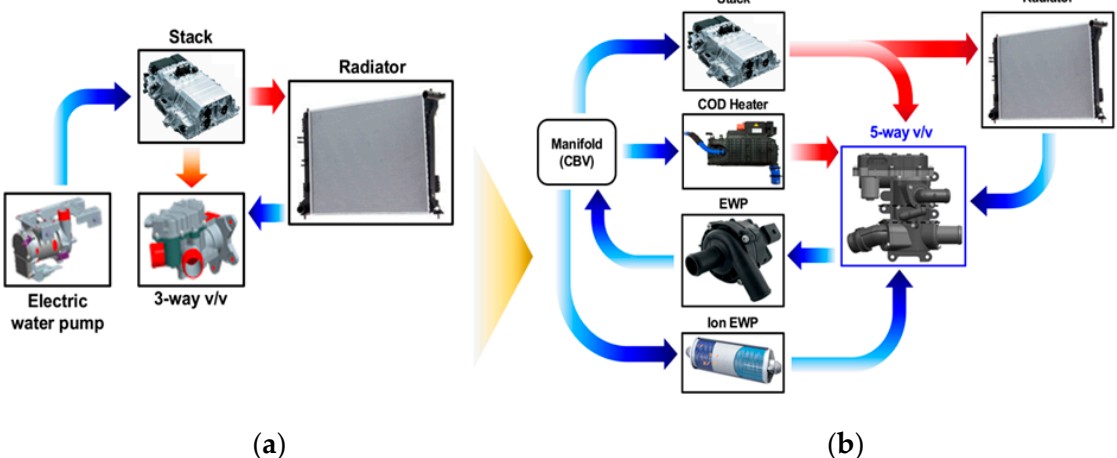

(**a**)  (**b**)

**Figure 2.** Illustration of five-way electric coolant control valve concept for an integrated module cooling strategy. (**a**) Three-way valve cooling circuit; (**b**) five-way valve cooling circuit.

As explained above, the electrical coolant valve is a critical component in the TMM as it modulates the amount of coolant flow to individual components in the cooling system, such as the stack, heater core, COD heater, and radiator. Therefore, optimizing the control strategy of the electric coolant control valve, which controls the electric valve's position under various vehicle operating conditions, is a significant design factor. However, system-level control logic optimization using actual vehicles has limitations because it requires considerable cost and time [9]. Above all, the optimizing control logic of electric coolant valves based on an actual vehicle has the disadvantage of not being able to optimize the control algorithm in the early stages of vehicle design.

Therefore, very recently, attention has turned to optimizing the thermal management system and corresponding control strategies using an integrated one-dimensional thermal management simulation model coupled with the thermal–hydraulic model, vehicle model, and controller model, with the understanding that these could offer a significant means to improve the vehicle's powertrain efficiency and fuel economy and the occupant's thermal

comfort [10–13]. The previous literature focused on various integrated thermal management system models with different control strategies based on fuel cell vehicles, which considered the cooling of the driving system, fuel cell stack, battery, and stack.

In most previous studies, several valves with junctions modeled the flow control valves, such as the mechanical thermostat, electric thermostat, and three-way electric coolant control valve, which controls the coolant flow rate and direction. The valve opening area was based on controlling the flow to different components. However, the model was built based on the assumption that the junction volume was small and incoming gases were well-mixed. Additionally, flow coefficients for each valve opening area that controlled the flow rate were calculated in steady-state conditions.

The five-way electric coolant control valve, which is the subject of research, has four inlets and one outlet, so the volume is large, and complex turbulent flow exists inside. Additionally, the ball valve rotates quickly to respond to various operating conditions, so the valve opens rapidly. Therefore, it is not easy to describe the movement of this five-way coolant control valve using conventional modeling techniques based on steady-state and well-mixed conditions. Ultimately, prediction accuracy will deteriorate if an integrated thermal management simulation model is constructed by modeling with the PCCV based on traditional modeling techniques.

Therefore, for the design and control of a five-way electric coolant control valve, it is not only necessary to understand the flow characteristics inside the valve but also to consider some complex flows (such as vortex, turbulence dissipation, and backflow) caused by fluid resistance during the spool opening process. To achieve this goal, there is difficulty in considering the fast rotation of the valve, which is the operating condition of the five-way valve. Accordingly, 3D dynamic flow characteristics research that simulates the rotation of a 3D ball valve has not been conducted due to obstacles such as high complexity, numerical instability, and high computational time.

Until now, most of the previous literature on internal flow characteristics considering the movement of the valve was geometrically simple, and the direction of the valve was also simple due to the difficulties mentioned above [14,15]. Recently, a numerical model based on the moving grid technology was carried out to study the effect of spool rotating speed on the internal flow characteristics of the cryogenic ball valve with one inlet and outlet [16].

In this study, to consider the dynamic inertia effect caused by ball valve rotation, the rotation of the ball valve, which is the internal rotating body of a five-way coolant control valve, was modeled three-dimensionally using the moving grid method. Using the model developed in this study, the dynamic behavior characteristics of the valve's internal flow, with respect to the ball valve's rotation speed, were studied. Dynamic flow characteristics have been quantitatively studied by comparing the flow rate and mixing temperature with the steady-state simulation results. This study's results will be helpful as primary design data for one-dimensional thermal–hydraulic modeling of a five-way valve to be applied to future integrated thermal management system simulation.

## 2. Problem Description

The geometrical and physical modeling involved in a five-way electric coolant control valve is complex and requires moving mesh approaches to account for the rotating valve movement based on three-dimensional compressible fluid dynamics.

The PCCV is responsible for regulating the coolant flow to the five components of the thermal management system (stack, COD heater, ion filter, radiator, and water pump) to control the fuel cell stack and maintain a temperature of 58 °C ($\pm$1 °C).

The structure of the five-way electric coolant control valve is shown in Figure 3 and mainly consists of the rotating valve, pipes to different components, housing, drive mechanisms, and electric circuits. The rotating valve controls the coolant flow to every element based on the layout of the pipes. The motor drives the rotation of the valve through the drive mechanism. The electric circuit provides power to the engine and measures the rotation angle for feedback.

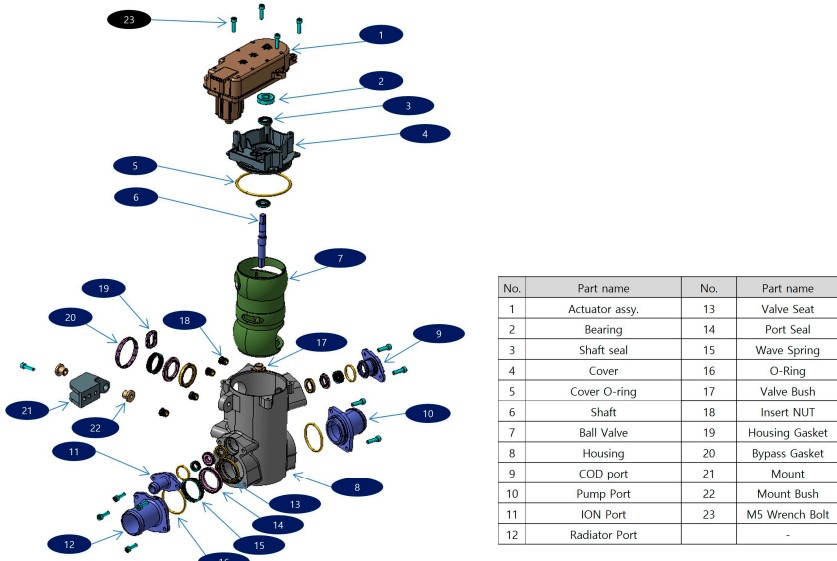

**Figure 3.** Exploded view of five-way electric coolant control valve.

Through the rotating motion of the PCCV's ball valve, the opening degree of the five ports connected to the thermal management system components is controlled, thereby generating primary pressure loss (orifice effect).

Figure 3 shows the part names and detailed diagrams of the five-way PCCV.

As shown in Figure 3, the rotating valve consists of three vertical layers and is connected to five ports. Therefore, a secondary loss (flow interaction loss) occurs in which the degree of opening and closing of the valve in each layer affects other layers. The flow streamlines shown in Figure 4 are when all ports of the PCCV are partially open and are the results of the flow simulations of this study. Ultimately, depending on the complex geometry of the rotating valve inside the PCCV, the turbulent streams flowing into the valve from each port interfere with each other, causing pressure loss and uneven enthalpy mixing due to turbulent viscous dissipation.

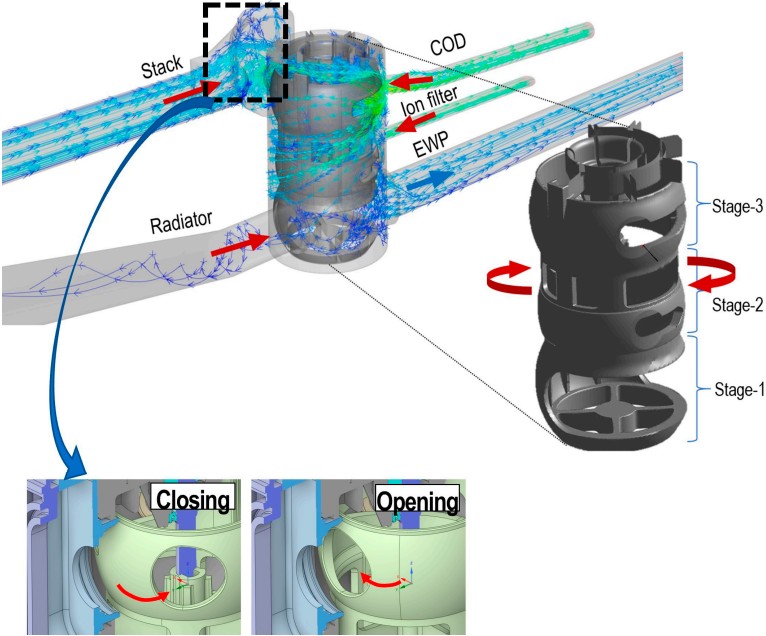

**Figure 4.** Geometrical details of five-way electric coolant control valve and internal flow paths.

## 3. Numerical Methods

To investigate the transient dynamic flow characteristics of the PCCV and quantitatively study the difference between dynamic and static flow characteristics, multiple CFD simulations of the flow were conducted using ANSYS Fluent [17] and Simerics-MP+ [18]. The three-dimensional steady-state flow analysis was performed using ANSYS Fluent, and the transient dynamic flow simulation using the moving grid method was calculated using Simerics-MP+.

In the following sub-sections, details of the simulation setup and results from the simulations are presented.

### 3.1. Computational Grid

The grid system, including the four inlet pipes, one outlet pipe, one rotating ball valve, and one valve housing, is shown in Figure 5. To allow relative motion between the internal rotating ball valve and the housing of the PCCV, there are clearances of a few microns between the rotating ball valve and the valve housing. However, the sealing action between the port gasket and the ball valve is assumed to be perfect in time, with a consequent null leakage flow in the gap between the rotating ball valve and the port inlet gasket. Therefore, this study neglected the clearance, and the support has no computational domain.

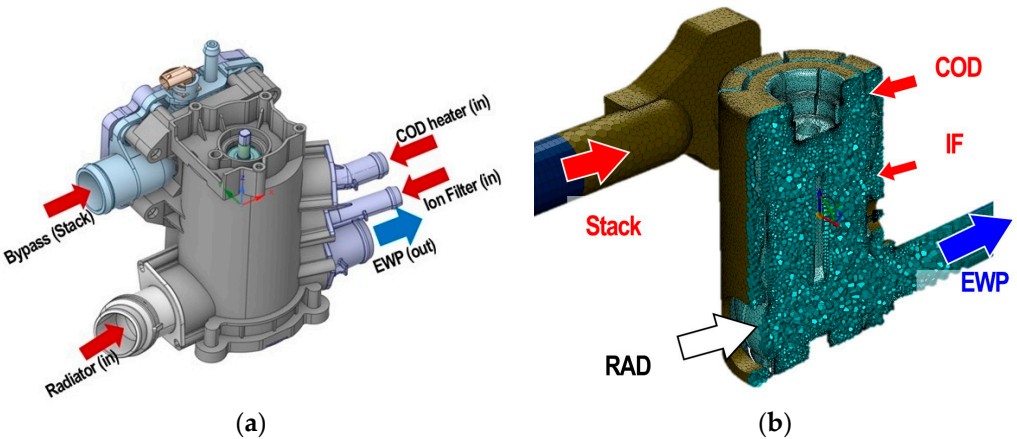

**Figure 5.** CAD surface data and computational grid system of the PCCV. (**a**) CAD surface data; (**b**) computational grid system.

To fully develop the flow of each inflow and outflow pipe, the pipes' length was 15 times the diameter of the pipes.

Figure 5 shows the Computer-Aided Design (CAD) surface data of the PCCV build-up by CATIA V5-6R2019) and grid system of computational domains. The fluid part of the PCCV is extracted from the CAD, provided by the industrial partner INZI CONTROLS, and the geometry is imported into CFD codes (Simerics-MP+ and ANSYS Fluent). The computational domains were discretized into meshes with structured and unstructured regions. As the flow conditions investigated in this work included turbulent flow regimes, special attention was given to the mesh size near the walls. In particular, as the wall functions were used in the turbulence modeling, the dimensionless distance of the first grid point from the wall, y+, was ensured to be $30 \leq y+ \leq 100$ [19]. In this study, the three-dimensional grid for considered geometry comprised 57,096 nodes, and 512,500 elements were imported to the flow solver.

### 3.2. Mathematical Formulation for Moving Grid

To study the effects of a ball valve rotating on the flow behavior inside the PCCV, dynamic flow simulations were conducted using the commercial CFD software Simerics-MP+ [18]. The authors prefer this software to ANSYS Fluent 2023 R1 [17] for transient dynamic flow simulation with moving/sliding grids because of its superiority in numerical

stability and ease-to-convergence [20,21]. Simerics software (ver. 5.2.15) has an advanced morphing capability, increasing computational efficiency and stability.

The entire flow region was split into two sub-regions for better computational efficiency. One was the ball valve area that rotated with the central axis, and the other was the non-deforming sub-region inside the valve housing that included four inlet ports and one outlet port. The movement of the rotating ball valve sub-region was controlled by the moving mesh algorithm embedded in CFD commercial software Simerics-MP+. Each sliding/moving cell is connected to the others through a standard interface, updated at each time step because of cell deformation and movement. The grid was explicitly remeshed, and the shape of the solution domain during the whole working process cooperated with the moving deforming boundary. The hexahedral structured grid and the moving mesh algorithm ensured the orthogonality of the deforming sub-domain while the orbiting scroll was rotating [18].

Governing equations for the moving grid technique of CFD are the mathematical expressions that describe the conservation of mass, momentum, and energy in a fluid flow with moving boundaries or interfaces. The governing equations for 3D compressible N-S equations can be referred to in [16,19]. The standard formulation of the conservation of the variable for a volume V with moving boundaries is based on the Reynolds transport theorem, which relates the rate of change of a conserved quantity inside a control volume to the flux of that quantity across the control surface. The general form of the conservation of the variable for a volume V with moving boundaries is:

$$\frac{\partial}{\partial t}\int_V \rho\phi dV = -\int_s \rho\phi(v - v_s)\cdot ndS + \int_V \rho\dot{\phi}dV \tag{1}$$

where $\rho$ is the density, $\phi$ is the variable of interest, $v$ is the fluid velocity, $v_s$ is the velocity of the control surface, $n$ is the unit normal vector pointing outward from the control surface, and $\dot{\phi}$ is the rate of production or destruction of $\phi$ per unit mass. Equation (1) can be applied to any conserved quantity, such as mass, momentum, or energy. The conservation of mass for a volume V with moving boundaries can be obtained by setting $\phi = 1$ and $\dot{\phi} = 0$ in the above equation, which gives:

$$\frac{\partial}{\partial t}\int_V \rho dV = -\int_s \rho(v - v_s)\cdot ndS \tag{2}$$

This equation states that the rate of change of mass inside the control volume is equal to the net mass flux across the control surface. Similarly, the conservation of linear momentum written in the moving boundary description is given as:

$$\frac{\partial}{\partial t}\int_V \rho v dV + \int_V [\rho v \otimes (v - v_s)]\cdot nds = \int_V \tau nds - \int_V pnds + \int_V \rho f dV \tag{3}$$

In this equation, $\tau$ is the Reynolds average viscous shear stress tensor, $p$ is the pressure, and $f$ is a body force which was neglected in this study. In this work, both turbulence and energy equations were considered. Further details concerning the moving grid technique and computational algorithm in software can be referred to in [14,18].

The opening and closing speeds of the PCCV were governed by setting the rotation speeds of the ball valve as 94 deg/s and 47 deg/s, respectively. These two different rotation speeds were obtained from experimental data originated by the driving vehicle test in the city and are the most frequent and representative rotation speeds. By setting the ball valve's rotation speed, each port's opening area changed continuously and the inflow mass flow rate was controlled accordingly. Figure 6 shows the opening area that changed as the ball valve rotated as a percentage for each inlet port. This means that 100% was fully opened. The *x*-axis represents the rotation angle. The angle at which the ball valve rotates clockwise from the closed position is defined as the rotating angle of the ball valve.

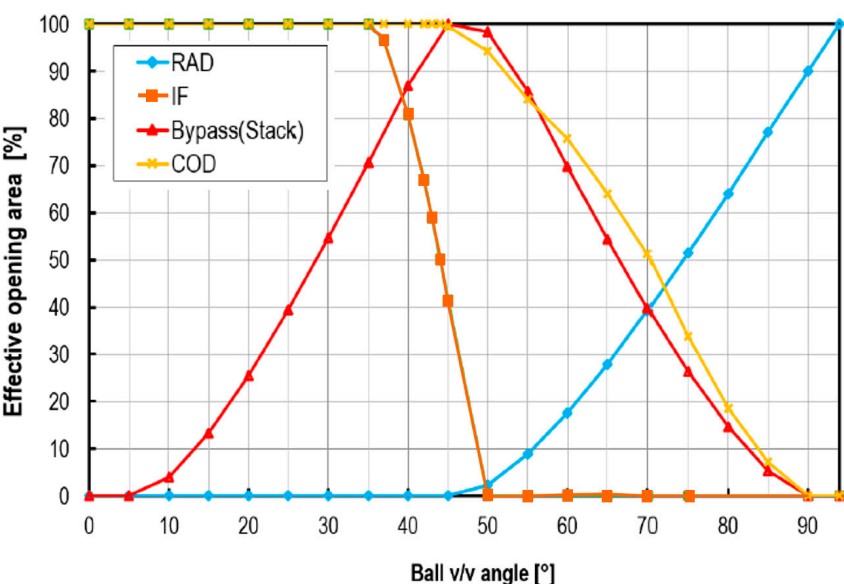

**Figure 6.** The variable opening area of each port with respect to the ball valve rotating angle (deg).

The rotation period of the ball valve was clockwise from 0° to 94° and then counter-clockwise from 0° to 94°, for a total of 188°, and the moving grid method was used to simulate the rotation of the ball valve.

### 3.3. Initial and Boundary Conditions

In this study, for stable transient dynamic calculation, steady-state analysis was performed with the condition of the ball valve at 0 degrees, and the result was used as the initial value in the transient dynamic calculation.

The working fluid was considered to be a half-mixture of ethylene glycol and water, whose density, thermal conductivity, specific heat, and dynamic viscosity were 1087 kg/m$^3$, 0.37 W/mK, 3285 J/kgK, and $9.07 \times 10^{-4}$ kg/ms, respectively. As shown in Figures 3 and 4, coolant streams cool the stack, COD heater, radiator, and ion filter, respectively, and flow into the PCCV through four inlet ports. Afterward, the coolant flow from each cooling component is mixed inside the PCCV and exits to the electric water pump through the outlet port.

Moving to the boundary conditions, pressure boundary conditions were applied at the inlet, and mass flow rate conditions were imposed at the outlet of the domain (as shown in Figure 5). No slip boundary condition was employed on the walls. The boundary conditions were obtained from experimental data from a test bench campaign held at the Department of Integrated Thermal Management System of INZI CONTROLS. For the boundary conditions, the inlet pressures at four inlet ports were atmospheric pressure, and two outlet mass flow rate conditions, 20 and 100 L per minute (LPM), were considered. The inlet thermal boundaries at inlet ports from the stack, COD heater, ion filter, and radiator were 75 °C, 30 °C, 30 °C, and 65 °C, respectively.

The steady-state calculation used the same boundary conditions as in the dynamic simulation case; the opening area at each port was imposed, as shown in Figure 6, and the ball valve was not moving.

### 3.4. CFD Solver and Numerical Implementation

CFD solvers used in this paper to perform the transient dynamic flow analysis of this PCCV were based on the finite volume method. They solved the conservation of mass and momentum equations along with turbulence and energy equations. As a finite volume method, the mesh of the entire fluid domain was discretized using high-quality structured and unstructured hexahedral cells.

The standard k-ε and the RNG k-ε models are the most common and mature turbulence models. Many others have different strengths and limitations depending on the flow problem, but these two models have been available for more than a decade and have been widely demonstrated to provide sound engineering results. The RNG k-epsilon model uses a modified value of $C_{\epsilon 2}$ included in the ε transport equation based on the Reynolds number and the turbulence intensity. Thereby, the RNG k-ε model is more accurate than the standard k-ε model in several applications, including fast strain flows, swirling flows, separated flows, and flows with strong streamline curvature, as the model constants are derived explicitly and contain a strain-dependent correction term [19]. Therefore, in this study, the RNG k-ε model [17–19] was selected as a turbulence model, considering its high accuracy and convergence stability optimized in the fast strain flow proposed by Yakhot and Orszag [22].

The iterative Semi-Implicit Method for Pressure-Linked Equations (SIMPLE) pressure–velocity coupling scheme was used to compute the pressure field. A second-order upwind scheme discretizes convective terms in momentum and energy equations. The convergence criteria were the reduction of scaled residuals [19] of continuity, x, y, and z momentum, k-ε equation, and energy equations below $10^{-6}$.

The computer used to calculate this study's transient dynamic CFD model was an AMD Ryzen Threadripper 3960× 24-Core Processor 3.79 GHz. When the rotation speed of the ball valve was 94 deg/s, the total calculation time was 5 h and 8 min; when the ball valve's rotation speed was 47 deg/s, it took 9 h and 50 min.

## 4. Simulation Results and Flow Analysis

In this study, we numerically studied the transient dynamic flow characteristics of a recently developed five-way electric coolant control valve to respond quickly to changing operating conditions and control coolant temperature precisely. For this purpose, CFD analysis was performed, considering the internal flow of the PCCV to be a three-dimensional compressible turbulent flow. In particular, the moving grid technique was adopted to simulate the rapid rotation of the ball valve accurately, as well as the flow interference and non-uniform mixing characteristics of streams flowing in from the four inlet ports with different temperatures and flow rates. To qualitatively study the dynamic flow characteristics of this PCCV, the results of the transient dynamic flow simulation were compared with the steady-state case where each ball valve did not rotate.

### 4.1. Transient Dynamic Flow Characteristics in Rotating Processes

The primary purpose of this study was to investigate the mutual interference and dynamic inertia effect of the jet-like incoming flow from each port due to the rotation of the ball valve. In Figures 7–10, the profiles of the flow rate flowing to the PCCV at each inlet face according to the rotation speed and direction of the ball valve are shown. The results of the transient dynamic simulation are compared to the steady-state results. The graphs on the right show the average error (difference) between the steady-state and dynamic results according to changes in the rotation direction and speed of the ball valve as bar graphs. The average error is defined by the equation as follows:

$$Averaged\ Error(\%) = \frac{\sum_{i=1}^{N}\left(\frac{|x_i - \overline{x_i}|}{\overline{x_i}}\right)}{N} \times 100 \tag{4}$$

where *i* is the number of a particular data value, *N* is the total number of data, $x_i$ is the dynamic mass flow rate, and $\overline{x_i}$ is the mass flow rate calculated from the steady state.

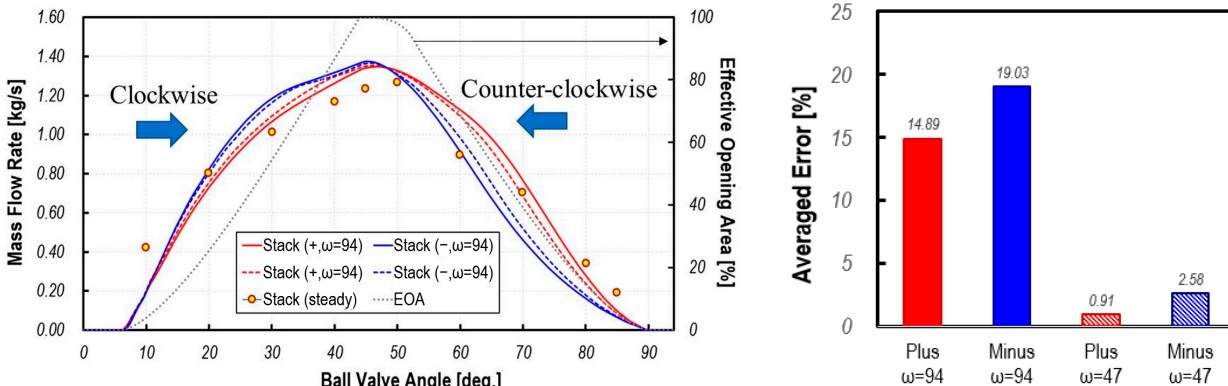

**Figure 7.** Temporal variation of the mass flow rate at the inlet face of the stack port for various rotation speeds and averaged discrepancies between transient dynamic flow analysis and steady-state analysis.

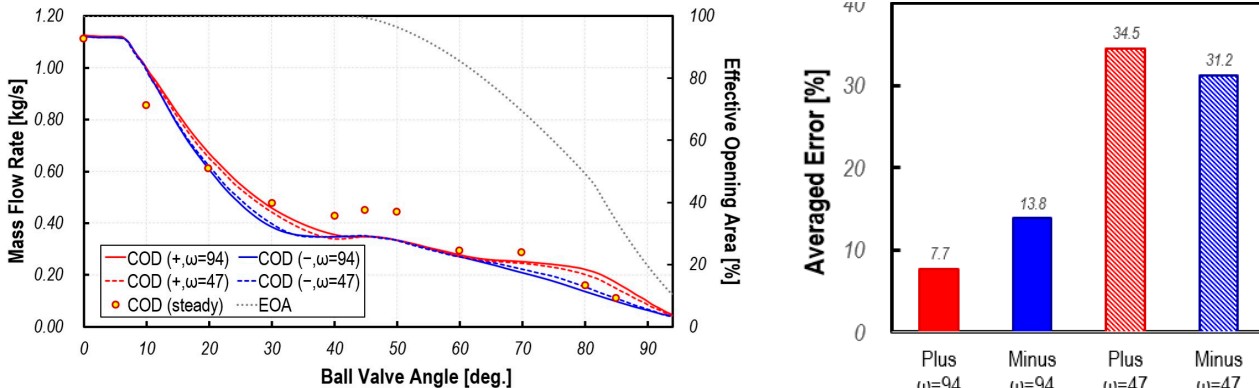

**Figure 8.** Temporal variation of the mass flow rate at the inlet face of the COD heater port for various rotation speeds and averaged discrepancies between transient dynamic flow analysis and steady-state analysis.

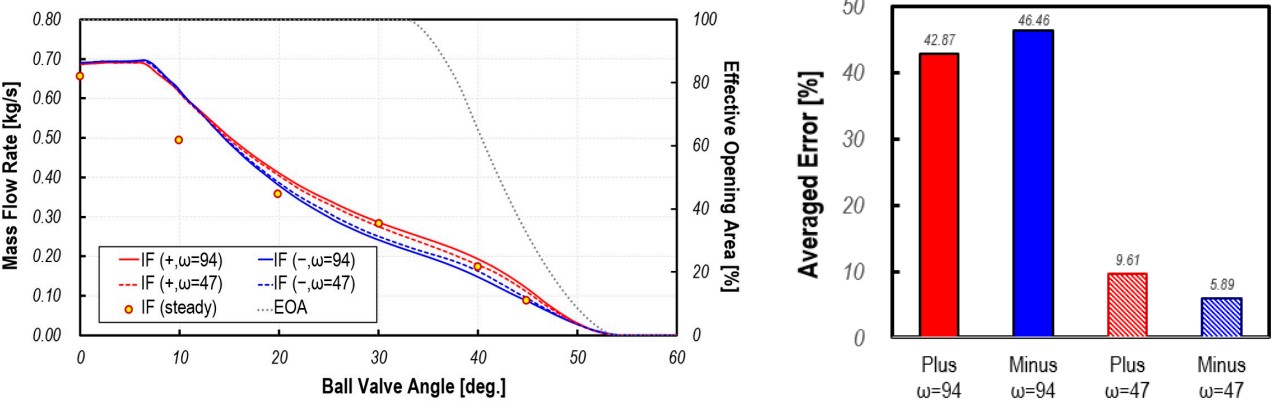

**Figure 9.** Temporal variation of the mass flow rate at the inlet face of the ion-filter port for various rotation speeds and averaged discrepancies between transient dynamic flow analysis and steady-state analysis.

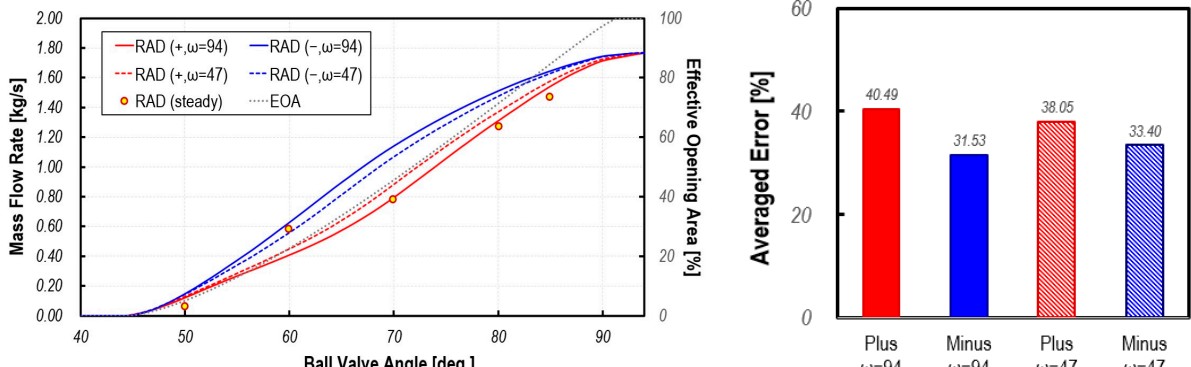

**Figure 10.** Temporal variation of the mass flow rate at the inlet face of the radiator port for various rotation speeds and averaged discrepancies between transient dynamic flow analysis and steady-state analysis.

In the case of the mass flow rate from the stack, as can be seen in Figure 7, under the condition of a high rotation speed of the ball valve ($\omega = 94$), the average difference in flow rate between dynamic simulation and steady-state simulation according to rotation is up to 19.3%, and under lower rotation speed ($\omega = 47$), there is a difference of up to 2.58%. Here, (+) means clockwise rotation and ($-$) indicates a counterclockwise process.

As can be seen from the results, the direction of rotation has a great effect on the dynamic mass flow profiles but the impact of rotation speed is negligible. In addition, many differences from the steady-state results occur near the opening and closing timing. In particular, the difference between the steady-state and dynamic analysis results increases as the closing time of the valve approaches (closing process). In the case of a high rotation speed ($\omega = \pm 94$), the difference in flow rate between the steady-state and dynamic simulations is up to 16%, and in the case of a slower ball valve rotation speed ($\omega = \pm 47$), the difference in flow rate is up to 8%. In particular, the most significant error can be found in ranges of 30 to 60 degrees, where the opening area of the stack port reaches its maximum.

Figure 8 compares the mass flow rate flowing into the PCCV through the COD heater with the results of rotation speed, direction, and steady-state calculation. A noteworthy feature is that the inflow rate continuously decreases in the 0–45 degree period, which is the fully open area. This is because the flow rate coming from the stack is gradually increasing in this section. As shown in Figure 5, this increase in flow from the stack causes interference with the inflow from the COD heater, ultimately leading to a gradual decrease in inflow. Afterward, as the inflow rate from the stack and valve opening area of the inlet port of the COD heater gradually decreases, the inlet mass flow rate from the COD heater gradually decreases. As shown in Figure 7, the rotation direction of the ball valve significantly affects the dynamic inlet mass flow profile, but the rotation speed does not. Under conditions where the ball valve rotation speed was high ($\omega = 94$), the average difference from the steady-state flow rate due to ball valve rotation was up to 13.8%. Under conditions where the ball valve rotation speed was slower ($\omega = 47$), the difference in flow rate between the steady-state calculation and dynamic simulation was as high as 34.5%. These results show that the temporal variation of the mass flow rate into the inlet port of the COD heater is more greatly affected by interference with the flow from the stack than by the dynamic inertia effect caused by ball valve rotation. Therefore, in the case of the PCCV with multiple inlet ports and a large internal volume, it is necessary to consider not only the rotation direction of the valve but also the mutual interference between jet-like flows flowing through each port into the PCCV to predict the accurate inlet flow rate.

Figure 9 compares the temporal variation of the flow rate from the ion filter into the PCCV with the steady-state calculation results for two kinds of rotation speeds and directions. In this illustration, the dynamic profile of the inflow mass flow rate is very similar to that of the COD heater port. This is because the ion-filter port is open when high-velocity jet-like streams flow from the COD heater port and stack port, resulting in

severe interference with these flows. Therefore, even when the inflow port from the ion filter is entirely open, the flow rate from the stack port gradually increases, and the flow rate from the ion-filter port decreases. From these results, it can be observed that control of the inflow rate by controlling open areas may become difficult due to interference with flows coming from other ports. Therefore, in the case of a multi-stage ball valve such as the PCCV, the opening strategy can be designed effectively when it is intended to minimize mutual interference from each port.

The difference in the mass flow rate between the clockwise and counterclockwise directions for two rotation speeds was up to 9% at $\omega = 94$ and 6% at $\omega = 47$. The average flow rate difference through the ion-filter port calculated from the steady-state and dynamic flow simulations was more significant than that of the COD heater and stack ports. This is because, in the case of the temporal variation of the mass flow rate of the ion-filter port, interference with flows coming from the stack port and COD heater port is more dominant than the dynamic inertia effect caused by the rotation of the ball valve.

Figure 10 shows temporal variations of the mass flow rate from the radiator as the ball valve rotates for two different rotating speeds and directions and these were compared with steady-state results. The flow rate of the inlet fluid flux from the radiator was higher than that of other ports, and the flow interference effect was minimal as the port was opened during the period when other ports were in the closing process. Therefore, it was observed that as the opening area of the radiator port increased, the inlet flow rate increased linearly.

The difference in flow rate according to the rotation direction was up to 40% at $\omega = 94$, which was not a significant difference compared to other cases. Still, under the condition of a slower ball valve rotating speed ($\omega = 47$), the difference in flow rate was up to 38.05%. The steady-state and dynamic simulation results showed a significant difference from other cases when the rotation speed was low because there was more flow interference with the inlet fluid flux of the three other ports. The difference between steady-state analysis and dynamic flow simulation results was 30–40% on average for $\omega = 94$ and 33–38% for $\omega = 47$, which was higher than other cases.

As can be seen from the results, since the momentum of the discharged inlet fluid flux was weak when the rotating speed was low, the inlet fluid flux from the radiator severely interfered with the inlet fluid fluxes from other ports. Since steady-state calculations could not consider the interference effects of complex inlet fluid fluxes and jet momentum flux changes due to the opening area's temporal variation, the difference between steady-state analysis and dynamic flow simulation appeared significant.

*4.2. Three-Dimensional Dynamic Inertia Flow Analysis in the PCCV*

In this section, the numerical results obtained from 3D dynamic flow simulation were analyzed to study the effect of ball valve rotation speed on the PCCV internal flow characteristics and were compared to those of steady-state solutions.

Figure 11 shows the velocity distribution due to the flow interaction of inlet fluxes from the stack and COD heater for different rotation directions at rotation angles, where $\theta = 40$ deg. As shown in Figure 11, the two inlet fluxes collided and flowed downward, and the flow rate from the COD heater was suppressed by the inlet flux with strong momentum flowing from the stack. This flow velocity distribution was also affected by the direction of rotation. When the ball valve rotates clockwise and reaches 40 degrees, the flow rate of flux flowing from the stack gradually increases, as shown in Figure 7. When it rotates counterclockwise, reaching 40 degrees right after the complete opening of the port opening area, the flow inertia is bigger and stronger than when it rotates clockwise. Therefore, a more robust jet-like stream from the stack and weaker inlet flux from the COD heater can be seen in the counterclockwise direction. It is observed that the results of steady-state flow calculations that cannot consider the flow inertia of the inlet fluxes underestimate the velocity and flow momentum of both the inlet flux from the COD heater and the inlet flux from the stack.

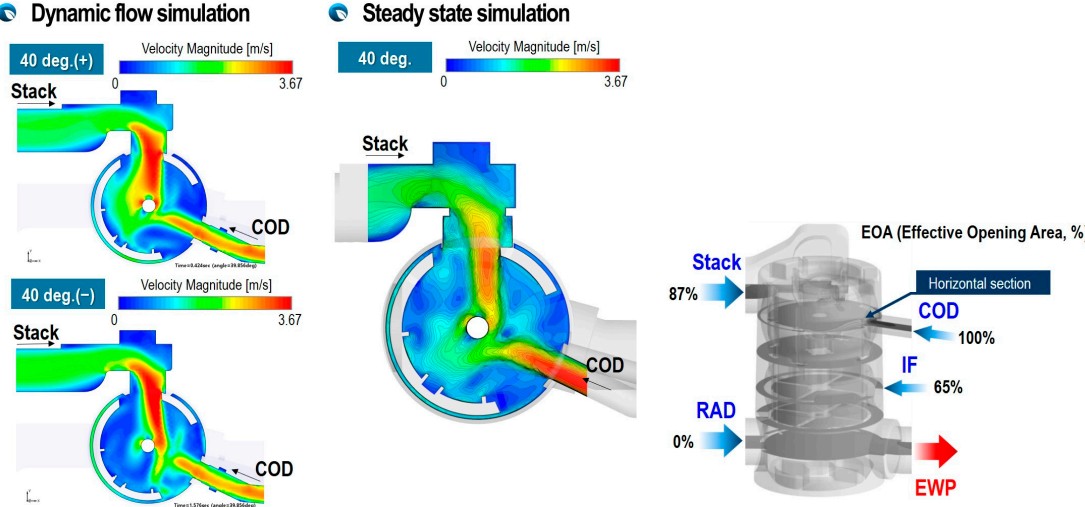

**Figure 11.** Cross-sectional view (@115 mm from bottom) of velocity distribution at $\omega = \pm 94$ deg/s and $\theta = 40$ deg.

Figure 12 shows the velocity distribution at the cross section just below the ion-filter port, where the jet-like flows coming in from three ports with fully open opening areas flow downstream with severe flow interference. A noteworthy feature is that when the rotating direction is clockwise, the incoming fluxes from the ion filter and COD heater, which have sufficient momentum, collide with the flux from the stack port and flow downstream. Therefore, the jet-like flows from the ion filter and COD heater are accelerating and moving downward as they hit the central wall of the plastic structure. However, in the counterclockwise direction, the flow inertia of the inlet flux from the ion filter and COD heater is not sufficiently developed, so it flows downward along the side wall of the plastic structure.

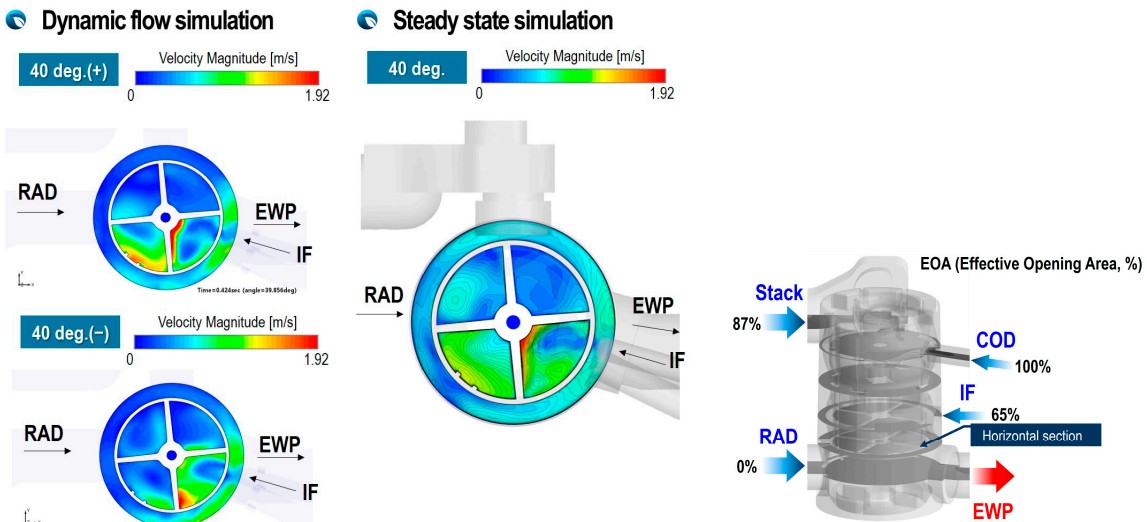

**Figure 12.** Cross-sectional view (@50.6 mm from bottom) of velocity distribution at $\omega = \pm 94$ deg/s and $\theta = 40$ deg.

The flow interference of inlet fluxes from the stack and COD heater at a rotation angle of 85 degrees is shown in Figure 13 for two rotation directions. It can be seen that the velocity distribution in this cross-section is different depending on the rotation direction of the ball valve. This is because, in the case of clockwise rotation, a rapid and robust flux from the stack port can be seen as the opening area of the stack port suddenly narrows. Still, in the case of counterclockwise rotation, the flow inertia of the flux from the stack is not

sufficiently developed and the inlet flux from the COD heater is very weak. However, it can be confirmed that the steady-state calculation results, which cannot reflect the time history experienced by the unsteady flow, cannot describe the strong jet flow that appears due to an instantaneous change in the opening area. Therefore, it can be seen that reflecting the flow inertia effect due to the rapid shift in the opening area and due to the rapid rotation of the ball valve is essential to accurately predict the flow distribution of the valve and the pressure decrease of the flow through the valve.

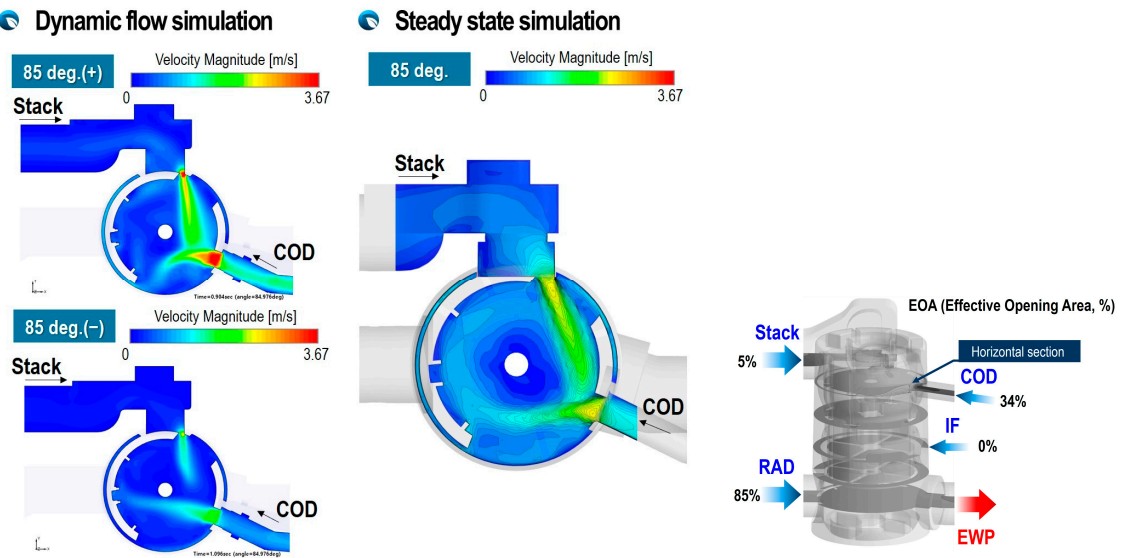

**Figure 13.** Cross-sectional view (@115 mm from bottom) of velocity distribution at ω = ±94 deg/s and θ = 85 deg.

Figure 14 shows the velocity distribution of the flux from the radiator passing through the inside of the PCCV according to the rotation direction of the ball valve. From the results, the case of counterclockwise rotation, where the flow inertia of the flux from the radiator port is extensively developed, is compared to the point of clockwise rotation, in which the flow inertia of the flux from the radiator port is not fully developed. Even though the opening area is the same, there are inlet fluxes with different momentum and flow inertia according to the rotation direction.

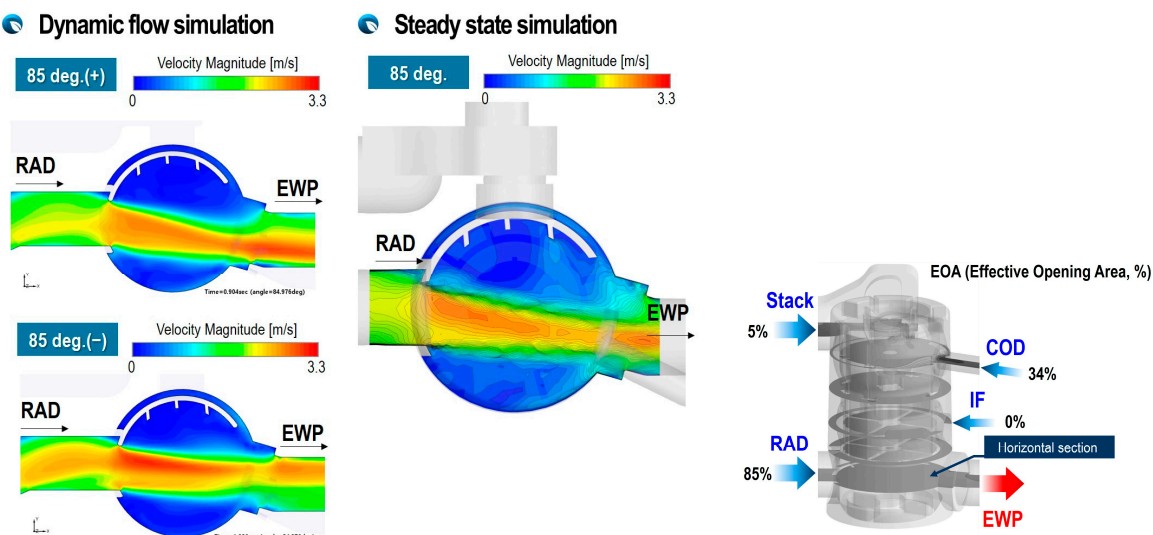

**Figure 14.** Cross-sectional view (@50.6 mm from bottom) of velocity distribution at ω = ±94 deg/s and θ = 85 deg.

Temporal sequential iso-views of particle streak lines in the PCCV are shown in Figure 15 to help understand the flow interference between fluxes flowing from each port. It can be seen that the jet-like stream discharged from the stack port strongly interferes with the fluxes flowing into the COD heater and ion-filter port. Consequently, as shown in Figures 8 and 9, a linear relationship between the inflow rate and the opening area is not established. The flow interference with jet-like streams shown in Figure 15 may cause a severe pressure decrease along the flow direction due to the viscous thermal dissipation converting the kinetic energy of the turbulent eddies into heat due to the viscous friction. As the rotation progresses and elapses, the flow rate of fluxes passing through the remaining ports except the radiator decreases, so the flux passing through the radiator port flows to the electric water pump without significant interference with other fluxes.

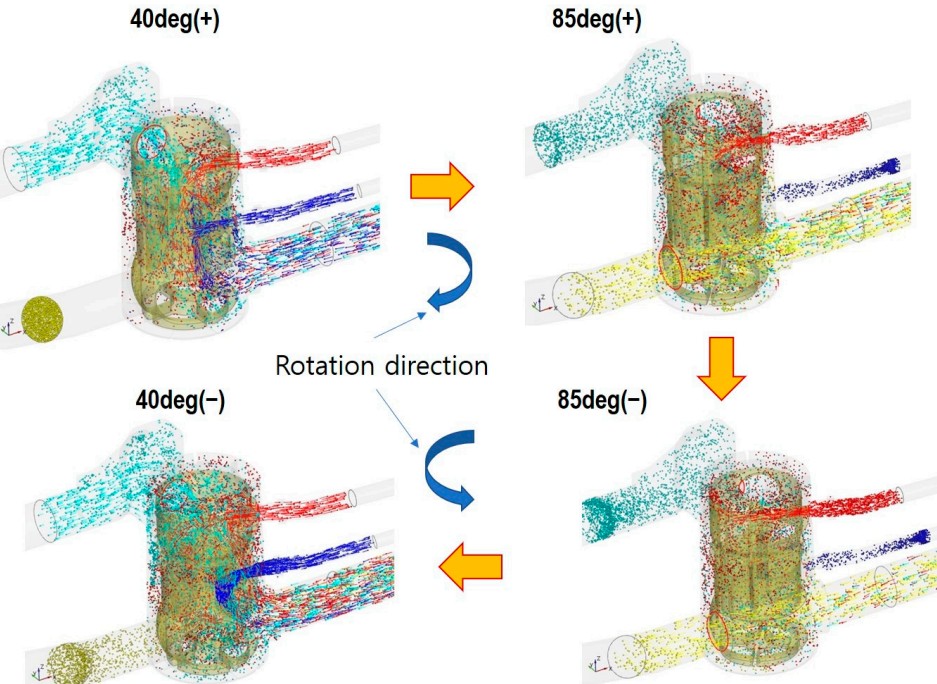

**Figure 15.** Temporal sequential isometric-views of particle streak lines in PCCV at $\omega = \pm 94$ deg/s (Color was used to distinguish flux flowing from each port).

Figure 16 shows the velocity pulsation profile at 3 mm inside the PCCV from the open face of the ball valve connected to each port. Here, Case 1 indicates a rotation speed of 94 deg/s and Case 2 indicates a rotation speed of 47 deg/s. It can be confirmed that very complex velocity pulsation profiles exist due to interference between jet-like streams with different momentum and flow inertia discharged from each port. Accumulated rotation angles represent the *x*-axis of graphs to display the complex pulsating velocity profile easily. The clockwise rotation range is from 0 to 94 degrees and the counterclockwise rotation range is from 95 to 188 degrees. First, the characteristics of the velocity pulsating profile inside the inlet of the stack port, which has the highest flow inertia among the fluxes of the three other inlet ports, are in sync with the opening area profile, as shown in Figure 6, because there is little interference with fluxes from other ports. In other words, a peak velocity profile is formed in the areas of 40–60 and 134–154 degrees of the rotation angle, which is near the maximum of the opening area. On the other hand, in the case of the COD heater ports and ion-filter ports, where the momentum and flow inertia of the incoming flux are relatively weak, they are not synchronized with the opening area profile due to severe flow interference with the flux from the stack with strong flow inertia and momentum. As shown in Figure 16b, the peak area in the velocity pulsation profile of the COD heater port occurs when the streamwise momentum and flow rate of the flux from the stack port are reduced. Moreover, in the case of flux from the ion filter, which has the smallest flow

inertia, the inflow velocity decreases sharply even in the fully open section of 0–40 degrees. This is because the momentum of the flux from the stack port increases steeply during this period, hindering incoming change from the ion-filter port. In the case of the radiator port, the inflow velocity increases rapidly and reaches its maximum in the rotation angle range of 60–90 degrees, where the opening area rapidly increases. This is because the distance from the radiator inlet port to the electric water pump outlet port is short, and the flow rate and flow inertia of the flux are large, so flow interference with other fluxes is small.

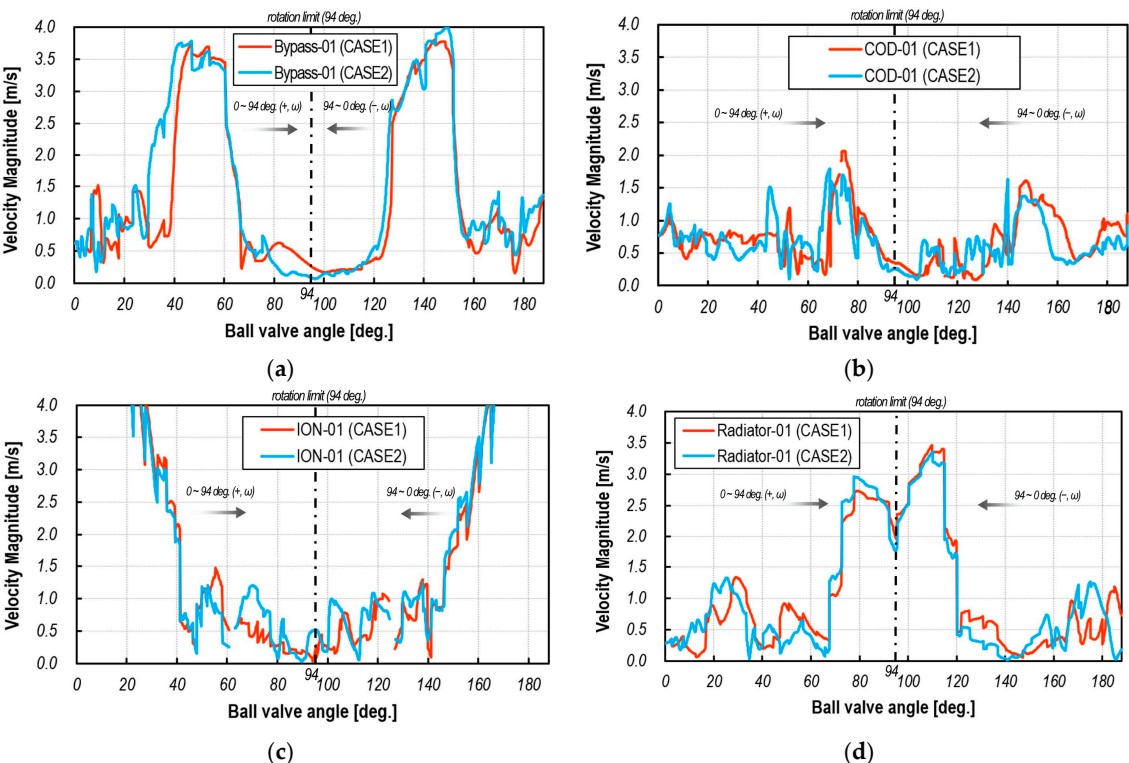

**Figure 16.** Temporal variation of velocity in PCCV (**a**) stack port; (**b**) COD heater port; (**c**) ion-filter port; (**d**) radiator port.

### 4.3. Transient Dynamic Thermal Mixing Characteristics in the PCCV

As explained above, the electrical coolant valve is a critical component in the TMM as it modulates the amount of coolant flow to individual components and, thereby, coolant temperature in the cooling system. Therefore, much of the previous literature [3–8] developed a one-dimensional integrated thermal management simulation model coupled with a thermal–hydraulic model to predict coolant temperature at several locations for coolant control strategy. However, in the case of the PCCV, which is the subject of this study, the control valve with a large internal volume and different levels of enthalpy streams from several inlet pipes. It is difficult to predict the outlet temperature accurately based on a simple calculation by weighted-average inlet enthalpies, as in Equation (5) below, which has been used in many previous one-dimensional models.

$$T_{mix} = f_{stk} \cdot T_{stk,out} + f_{cod} \cdot T_{cod,out} + f_{if} \cdot T_{if,out} + f_{rad,out} \begin{cases} 1 = & f_{stk} + f_{cod} + f_{if} + f_{rad} \\ f_{stk} & \text{(stack mass fraction)} \\ f_{cod} & \text{(COD mass fraction)} \\ f_{if} & \text{(IF mass fraction)} \\ f_{rad} & \text{(RAD mass fraction)} \\ T_{stk} & \text{(stack outlet temperature, }^\circ\text{C)} \\ T_{cod} & \text{(COD outlet temperature, }^\circ\text{C)} \\ T_{if} & \text{(IF outlet temperature, }^\circ\text{C)} \\ T_{rad} & \text{(RAD outlet temperature, }^\circ\text{C)} \end{cases} \quad (5)$$

Therefore, to analyze the effect of dynamic inertia flow characteristics on thermal mixing inside the PCCV and quantify the error in this study, the outlet temperature by dynamic flow simulation using moving grids was compared with the results from steady-state simulation and Equation (5).

Figures 17–20 show the temperature distribution inside the PCCV at $\omega = \pm 94$ deg/s at various angles in cross-sections at several axial locations. Summarizing the results, it can be noted that even at the same rotation angle, the temperature distribution is different according to the rotation direction of the valve. This is because the flow inertia and momentum of each flux are different. After all, the time history of boundary conditions experienced by each inlet flux differs because there are multiple inlet ports, and the enthalpy, momentum, and flow inertia of the inlet fluxes from them differ. Therefore, in the case of the PCCV, extremely non-uniform thermal mixing occurs due to flow interference, making it challenging to predict the mixing temperature accurately at the outlet using a simple average method, such as Equation (5).

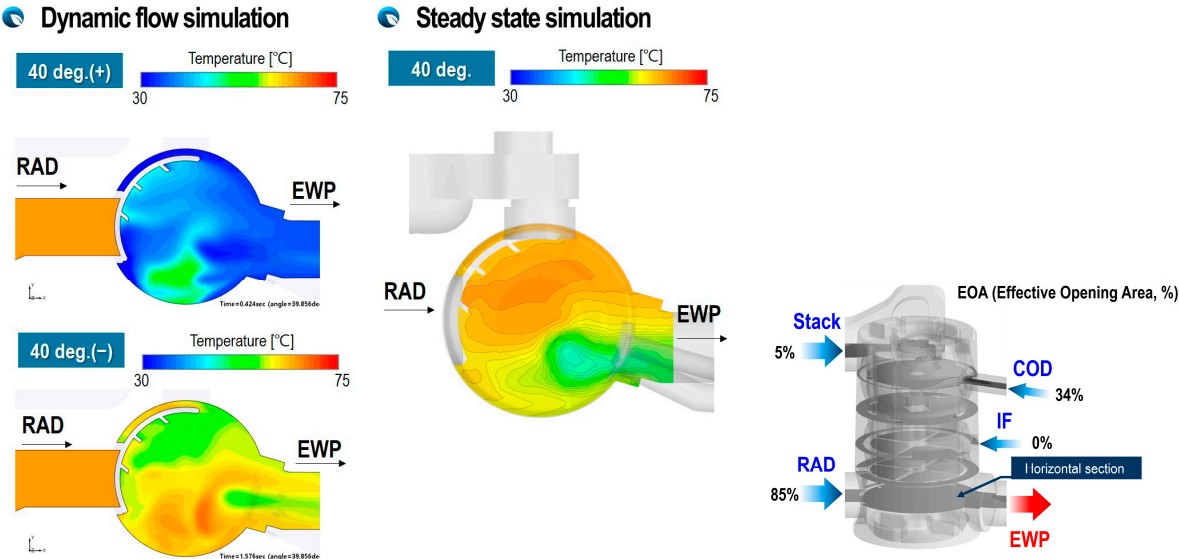

**Figure 17.** Cross-sectional view (@25.6 mm from bottom) of temperature distribution at $\omega = \pm 94$ deg/s and $\theta = 40$ deg.

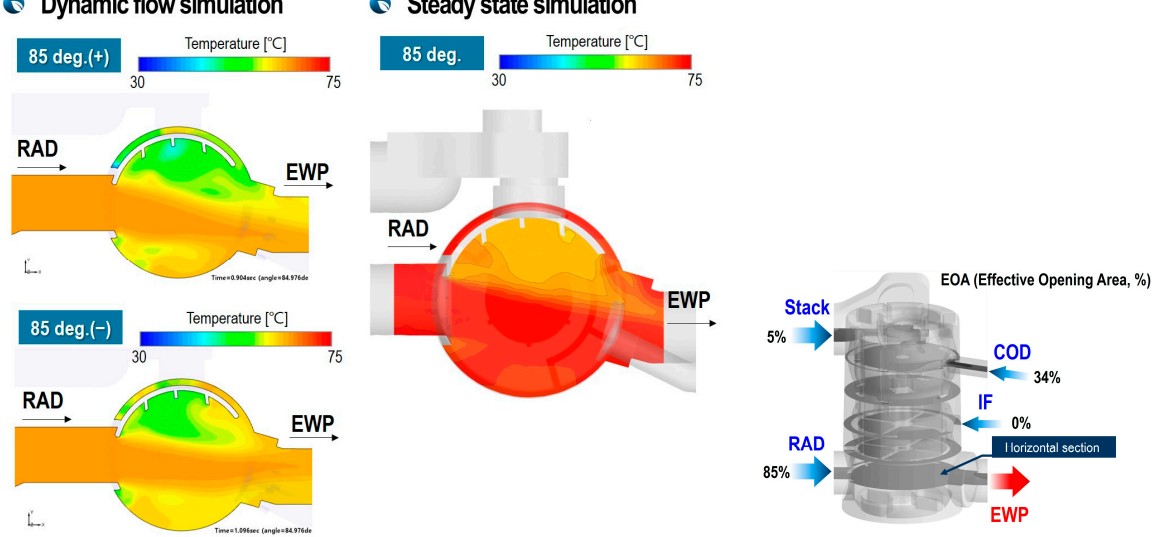

**Figure 18.** Cross-sectional view (@25.6 mm from bottom) of temperature distribution at $\omega = \pm 94$ deg/s and $\theta = 85$ deg.

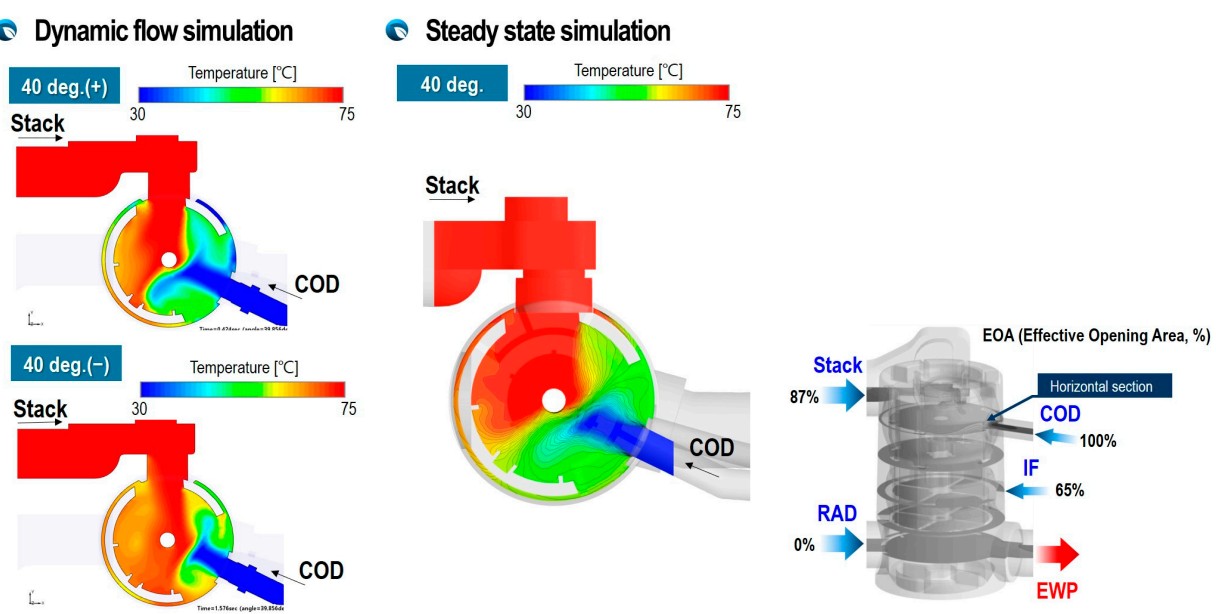

**Figure 19.** Cross-sectional view (@115 mm from bottom) of temperature distribution at ω = ±94 deg/s and θ = 40 deg.

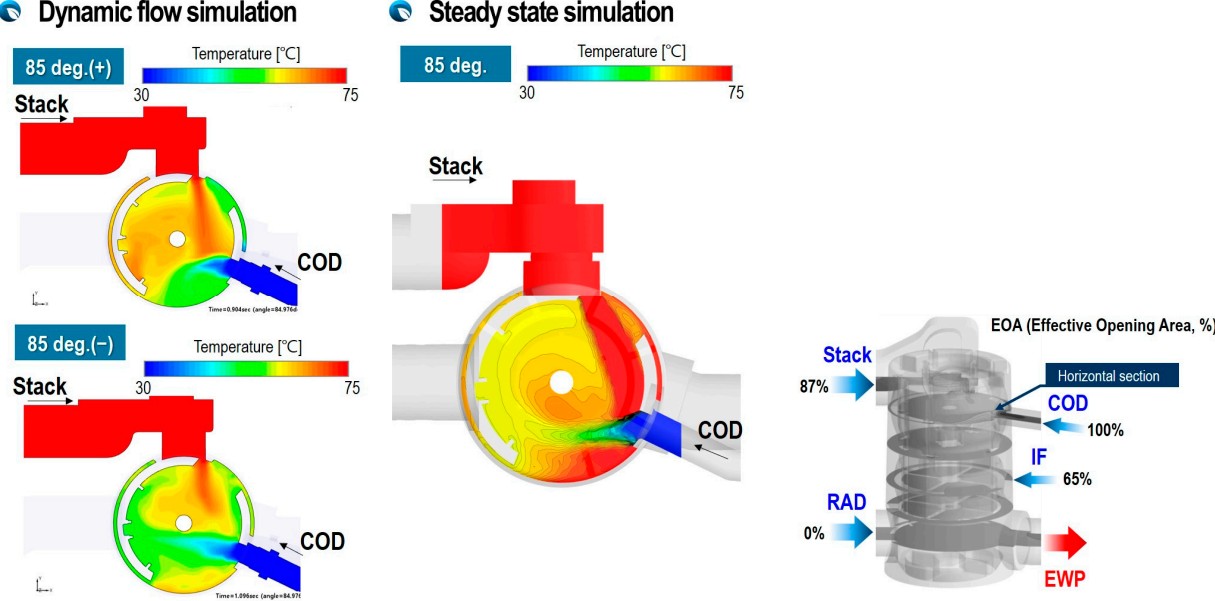

**Figure 20.** Cross-sectional view (@115 mm from bottom) of temperature distribution at ω = ±94 deg/s and θ = 85 deg.

Figures 21 and 22 quantitatively compare the mixing temperatures of the flux flowing to the electric water pump according to the rotation speed and direction at ball valve angles of 40 and 80 degrees. Additionally, the mixing temperature (denoted as theory) calculated from Equation (5) was also compared. The graphs on the right of each figure show the temperature difference from the mixing temperature calculated for a steady-state condition.

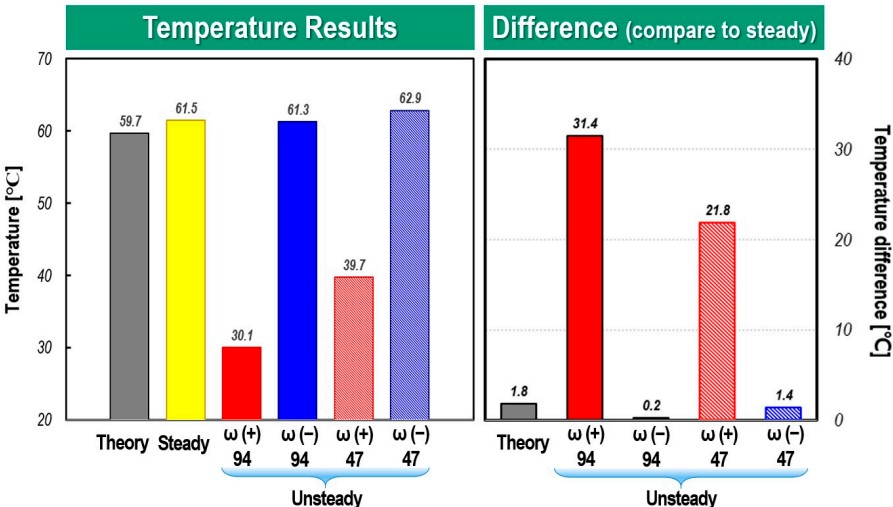

**Figure 21.** Comparison of temperature at the outlet between dynamic and steady-flow simulations for two different rotation speeds and angles at θ = 40 deg.

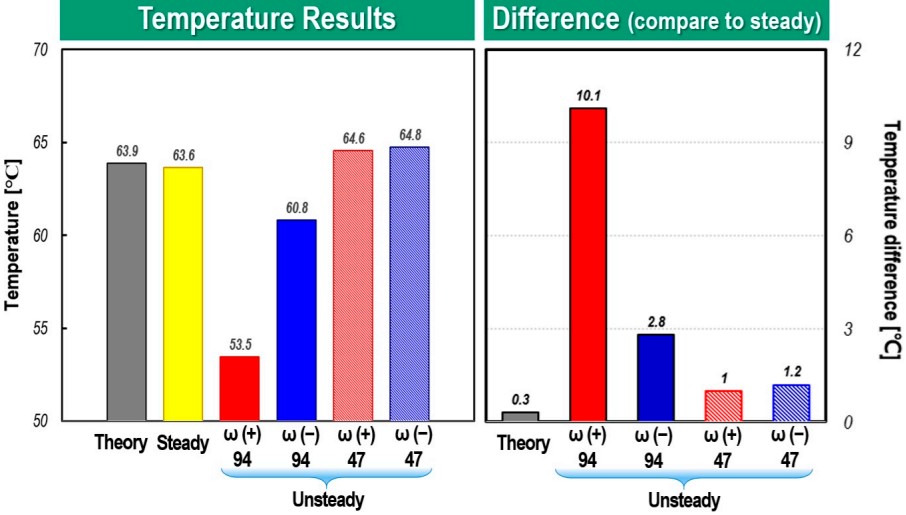

**Figure 22.** Comparison of temperature at the outlet between dynamic and steady-flow simulations for two different rotation speeds and angles at θ = 85 deg.

In the case of a rotation angle of 40 degrees, where there is severe interference between the fluxes discharged from the COD heater and stack port, and particularly in the case of a fast rotation speed of ω = +94 deg/s, the temperature difference from the steady-state result is up to 31.4 °C. On the other hand, in the case of ω = −94 deg/s, there is only a 0.2 °C temperature difference from the steady-state result even though the rotation speed is the same. This is because in the case of clockwise rotation, where the flow inertia of the fluxes from the COD heater and the IF is sufficiently developed, the high temperature flux coming from the stack is sufficiently cooled, but in the case of counterclockwise rotation, the cooling by these fluxes is relatively weak. Therefore, the mixing temperature of ω = −94 deg/s is significantly lowered compared to that of ω = +94 deg/s, resulting in a large mixing temperature difference from the steady-state calculation.

In the case of ω = ±47 deg/s, the maximum temperature difference was up to 21.8 °C. From the results, it can be seen that when the rotation speed of the ball valve is high, the temperature mixing inside the PCCV is highly non-uniform due to different flow inertia and flow interference between the incoming fluxes. What is more noteworthy is that the flow inertia from each port changes depending on the direction of rotation, greatly affecting thermal mixing. The difference between the mixing temperature calculated from the steady-

state analysis and Equation (5) was only 1.8 °C. However, the steady-state model with no consideration of rotation speed and direction shows a maximum 31.4 °C difference in the mixing temperature compared to that of dynamic flow simulation. Therefore, it can be deduced that if a cooling circuit using a multi-way coolant control valve, such as the PCCV, is modeled one-dimensionally, many errors are expected to occur in mixing temperature predictions and control logic optimization. As can be seen in Figure 22, in the case of a rotation angle of 85 degrees with weaker flow interference, it is evident that the dynamic flow effect is reduced. However, the dynamic behavior effect is shown clearly in the case of a high rotation speed.

## 5. Conclusions and Remarks

In this study, the three-dimensional thermo-fluid CFD model of the five-way electric coolant control valve (PCCV) considered flow inertia and dynamic flow characteristics. To achieve this goal, moving grid techniques considered flow inertia and dynamic flow in the opening and closing stages of the ball valve rotating motion.

The simulation results demonstrate that three-dimensional dynamic inertia flow significantly impacts flow distribution and thermal mixing in a PCCV. In other words, as fluxes with different levels of enthalpy and momentum flow into the PCCV, their flow interference and inertia make dynamic flow characteristics stream and thermal mixing patterns utterly different from the steady-state solution. This study found that the dynamic flow and thermal mixing characteristics inside the PCCV were greatly affected by the rotation speed, rotation direction, and degree of flow interference between fluxes.

The most influential factor representing dynamic flow characteristics is the rotation direction of the ball valve. In addition, it was found that if there was strong interference with fluxes flowing in from other ports, the difference of the inflow rate between the dynamic and steady-state calculations became larger. In particular, it was observed that the average difference of the inflow rate through the ion-filter port between the dynamic and steady-state calculations was up to 46.46% due to the dynamic flow characteristics, strong flow interference, and rapid rotation of the ball valve. Therefore, in order to determine the optimal transient profiles of the opening area for each inlet port of the PCCV, it is essential to consider the dynamic flow characteristics inside the PCCV, as shown in the results of this study. The transient dynamic flow of the PCCV also has a strong effect on the thermal mixing pattern. Therefore, it was confirmed that the mixing temperature of the flux to the outlet changed significantly depending on the rotation direction and speed of the ball valve.

This study can be also used as a reference for one-dimensional hydraulic modeling of a PCCV, which can consider the effect of the flow inertia, flow interference, and corresponding integral thermal management framework, helping to achieve better coolant control strategies in the future.

**Author Contributions:** Conceptualization, S.-J.J. and J.-h.K.; methodology, S.-J.J.; software, S.-J.J. and S.-J.M.; validation, S.-J.J.; formal analysis, S.-J.J., S.-J.M. and G.-s.L.; investigation, S.-J.J.; resources, S.-J.J., J.-h.K., S.-J.M. and G.-s.L.; data curation, S.-J.J. and J.-h.K.; writing—original draft preparation, S.-J.J.; writing—review and editing, S.-J.J.; visualization, S.-J.J. and S.-J.M.; supervision, S.-J.J.; project administration, S.-J.J. and S.-J.M.; funding acquisition, S.-J.J. and J.-h.K. All authors have read and agreed to the published version of this manuscript.

**Funding:** This work was supported by the core technology development for the new industry entry business reorganization Project of the Korea Institute for Advancement of Technology, funded by the Korean government (Grant No. P0017348).

**Data Availability Statement:** Most of the data presented in this study are available on request from the corresponding author.

**Conflicts of Interest:** Ji-hoon Kang and Gum-su Lee are from INZI CONTROLS Co., Ltd., all the authors declare no conflicts of interest.

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
