# Peer review of "Transient and Dynamic Simulation of the Fluid Flow through Five-Way Electric Coolant Control Valve of a 100 kW Fuel Cell Vehicle by CFD with Moving Grid Technique"

_actuators, doi:10.3390/act13030110_

Round 1

Reviewer 1 Report

Comments and Suggestions for Authors

No major comments. Overall the paper is structured well and clearly guides the reader to the conclusions made. The introduction, design approach and discussion of CFD methods are good. The main area for imorvement is to expand on the conclusions. Given the extent of the results/discussion, it would be expected to have a larger conclusions and perhaps more detail on the link to future work. 

Author Response

See the attached file, please.

Reviewer 2 Report

Comments and Suggestions for Authors

Dear Authors,

the manuscript investigates the transient dynamic fluid flow characteristics of a 5-way electric coolant valve through three-dimensional dynamic fluid flow simulation through the valve using a commercial CFD solver with a moving mesh technique. The results are new and important from a scientific and application point of view. The work is interesting and worth publishing in the journal with some corrections.

Detailed comments on the manuscript:

1. Too much keywords.

2. I think it is worth supplementing the literature because there are too few references in the manuscript. I propose to add the following works regarding PEM fuel cell: https://doi.org/10.4271/2007-01-2012
and hydrogen fuel cell: doi: 10.26552/com.C.2021.4.E56-E67
and in the field of flow modeling: https://doi.org/10.4271/2014-01-2883

3. Check the correctness of the drawing signature with the template.

4. Move Figure 3 to line 193.

5. The work arrangement should be changed slightly. I suggest changing section 2 to Methodology, then: 2.1. problem descriptions; 2.2. Numerical Methods. 4. Mark simulation results and flow analysis as section 3.

6. Move Figure 5 to line 224.

7. Move Figure 6 to line 301.

8. Move Figure 9 to line 437.

9. In line 449, the description concerns the later part, i.e. Fig. 10. Move Figure 10 to line 455.

10. Move Figure 10 to line 492.

11. Move Figure 16 to line 558.

12. Move the sentence "Figures 21 and 22 quantitatively compare the mixing temperatures according to each analysis method." to line 646. then Figure 22 to line 652, under Figure 21.

13. The conclusions are synthetic and correct.

14. There are also minor editing errors, e.g. lines 12, 38, 41, 247, 249, 265, 292, 326, 351.

Thank you

Reviewer 3 Report

Comments and Suggestions for Authors

The paper focuses on the CFD analysis of a 5 way valve in fast opening and closing conditions: the authors emphasize the difference of the valve behavior (i.e., mass flow and coolant temperature) between steady state and transient conditions.

The subject is relevant, the methodology is clear and the results are interesting. However, the quality of English language a should be improved: a proof reading is highly recommended. Furthermore, many repetitions should be avoided, and the presentation of the results should be clearer.

One general remark concerning the approach proposed by the authors: in the abstract they state that the objective is to improve the quality of coolant temperature control strategies, but after reading the paper it is not clear how the transients they take into consideration (constant speed of 47/94 deg/s) significantly represent an actual use case. While the authors compare the mass flow for given ball valve angle in steady state and dynamic (constant speed) conditions , it would also be interesting to assess the time required to reach steady state results, after a given position of the ball valve has been reached, both in terms of flow and temperature. From the  point of view of the coolant temperature control, tests like step variations of the ball valve angle would be more significant. Large differences in mass flow and temperature may not be significant if the duration of the discrepancy is infinitesimal: the authors should add elements to support their choice of simulating the valve behavior at constant speed, or they could add a simulation to demonstrate that the same results would be obtained for an actual use case of the valve.

A few specific suggestions to improve the quality of the paper:

- The components numbers in Figure 3 are not clearly visible    

- The unit of measurement of kinematic viscosity reported in section 3.3 (kg/ms) is actually that of dynamic viscosity. Please, also check the value. 

- In the discussion of achieved results, some parts should be clarified: for example, in the comment of Figure 7 the authors state: '...the direction of rotation has little effect on the dynamic mass flow profiles but the rotation speed impact greatly.'. Looking at figure 7, it appears that the blue lines (negative speed) referring to different speed (47 and 94 deg/s) are much more grouped, with respect to those referring to the same speed, but different direction of rotation (red lines, positive speed, 47 and 94 deg/s). 

- It is also difficult to interpret the bar graphs showing the average error: referring once again to figure 7, the trends for positive speed 47 and 94 deg/s are quite similar, and the distance from the steady state points in the left plot (i.e., what is called 'error' in the right plot) should be similar. However, average errors are significantly different (0.91 and 14.89, respectively). The same applies to the following figures (8, 9): it is hard to understand why very similar mass flow rate trends end up in very different errors.

My suggestion is to clarify how the average error (a definition would help) is related to the mass flow rates trends and their difference with steady state points reported in the left plots.

- In line 449 the authors wrongly refer to Figure 9 (should be Figure 10).

- the use of tables collecting the main results would simplify the discussion, avoiding several repetitions: for example, in section 4.1 the authors repeat the same comments/concepts several times, to illustrate results achieved for different ports of the valve. The use of a table may help summarizing overall results. The same may apply in the following sections, where results referring to different conditions/sections of the valve are described.

- Figures 11,12 and 13,14 report velocity distributions referring to different cross sections: the information of the given cross section should be reported in the caption of the figure. The same applies for Figures 17,18 and 19,20

- In Figure 20 the valve 3-D sketch is reported in the bottom, while in Figures 17, 18, 19 it is reported on the right. Please, change Figure 20 accordingly.   

-  I suggest to avoid using percentage errors in Figure 21 and 22: it does not make sense on a temperature (especially if measured in °C). The simple temperature difference would be much more significant. In the same figure, the caption 'theory' should be clarified (stating that it refers to the evaluation carried out using eq. 4). The legend may be removed form these figures, as captions are reported on the x-axis.

Comments on the Quality of English Language

The paper requires a proof reading: several sentences should be rephrased, verbs tenses and singular/plural agreements should be checked.

Round 2

Reviewer 2 Report

Comments and Suggestions for Authors

Dear Authors,

thank you for making changes to the manuscript and responding to the review. I accept the answers. The article has been significantly revised according to the suggestions from the review, and the literature has been supplemented. There are a few editing errors (e.g.: 47deg/s or 1.8℃, there should be a space between the number and the unit), which should be corrected in the final version of the manuscript. Furthermore, I noticed that the list of authors of ref. [1] is incorrect, it should be: Wendeker, M.; Malek, A.; Czarnigowski, J.; Taccani, R.; Boulet, P.; Breaban, F.

Due to the above, I recommend the article to be printed with minor corrections.

Thank you!
